# Improvement of Selected Quality and Safety Traits in Turmeric-Enriched Kale Pesto Using Blue Light and Sous-Vide

**DOI:** 10.3390/molecules29245831

**Published:** 2024-12-11

**Authors:** Magdalena A. Olszewska, Anna Draszanowska, Aleksandra Zimińska, Małgorzata Starowicz

**Affiliations:** 1Department of Food Microbiology, Meat Technology and Chemistry, The Faculty of Food Science, University of Warmia and Mazury in Olsztyn, Plac Cieszyński 1, 10-726 Olsztyn, Poland; 2Department of Human Nutrition, The Faculty of Food Science, University of Warmia and Mazury in Olsztyn, Słoneczna 45F, 10-718 Olsztyn, Poland; 3Department of Chemistry and Biodynamics of Food, Institute of Animal Reproduction and Food Research of Polish Academy of Sciences, Juliana Tuwima 10, 10-748 Olsztyn, Poland; m.starowicz@pan.olsztyn.pl

**Keywords:** blue light, sous-vide, polyphenols, antioxidant activity, color, *Listeria monocytogenes*

## Abstract

The potential of blue light (BL) and sous-vide (S-V) as a novel approach for food preservation was investigated via measurements of the total phenolic content (TPC), antioxidative activity, color, and their antibacterial effect on *Listeria monocytogenes* in two versions of laboratory-prepared kale pesto, with and without the addition of turmeric. The TPC ranged from 85 to 208 mg/100 g GAE d.m. and 57 to 171 mg/100 g GAE d.m., respectively. In both versions, the highest TPC was in the blue light–sous-vide samples, while the lowest was after the sous-vide, with a loss of polyphenols of almost 40% during storage when turmeric was absent. Antioxidative capabilities of the pesto were initially estimated at 54.07 and 7.46 µmol TE/g d.m., respectively, indicating significant bioactivity enhancement by turmeric. In turmeric-enriched pesto, sous-vide decreased the antioxidative activity levels by 12% in fresh pesto and by 45% during storage. Meanwhile, blue light compensated for the losses caused by the sous-vide treatment. Although the hue angle (*h*°) of sous-vide pesto was lower than that of blue light pesto in most samples, sequential BL and S-V ultimately yielded the lowest *h*°. The sequential BL and S-V treatment resulted in a 1.7 log reduction in the *L. monocytogenes* population, whereas adding turmeric increased the treatment efficacy by another 2.0 logs. Thus, as a source of photosensitizing molecules, turmeric was highly antibacterial after photothermal activation with blue light and sous-vide. This study suggests that blue light could be an effective (pre)treatment used on pesto sauces to preserve bioactivity and to improve safety when enriched with a natural additive like turmeric.

## 1. Introduction

Growing consumer demand for food ingredients with biological activity has significantly increased the popularity of leafy vegetables [1]. They can be utilized to prepare dressings, salads, sauces, and whole meals [2]. While, for instance, lettuce is solely distributed for fresh consumption, many other varieties, like kale, broccoli, and spinach, can be utilized in processed foods [1]. Pesto is an example of the latter, with a large selection of pesto sauces available on the market. Although it is traditionally made with basil, several improvements in pesto preparation have been made; for instance, with coriander as the main ingredient [2]. Given that large quantities of plant material are used in the preparation of pesto, consumers could appreciate a kale green pesto variety due to its color, taste, and rich source of bioactive compounds [3,4]. Kale was found to have the second strongest antioxidative activity among twenty-two common vegetables [5]. Its antioxidative and free-radical scavenging activities are derived from various components, with phenolic compounds being among the most important [6].

The microbiological stability of food products, including pesto sauces, is ensured through methods such as pH reduction, pasteurization, lowering of the water activity (a_w_), refrigeration, the use of modified atmosphere packaging (MAP), and combinations of these [7]. Despite their effectiveness, they often destroy valuable ingredients such as heat-sensitive vitamins and polyphenols [8]. Alternative preservation procedures have been developed to ensure the stability and viability of bioactive molecules to meet the growing demand for high-quality foods like pesto [9]. In particular, non-thermal technologies have minimal negative effects on food when applied as a pretreatment through moisture elimination, cell permeability modification, and improvements to bioactive properties [10]. That is why current research combines well-regarded processing technologies with natural antioxidants or innovative non-thermal technologies, resulting in beneficial effects on both quality and safety issues [11].

Among the non-thermal technologies, light-based irradiation systems, particularly light-emitting diodes (LEDs) have received increased attention due to their efficiency, safety, and potential for food preservation [12]. Blue LEDs with wavelengths of 405, 415, and 425 nm, in particular, have gained a reputation as a novel technology for bacterial inactivation [13]. Blue-light treatment involves the photoactivation of endogenous or exogenous molecules to generate reactive oxygen species (ROS), resulting in the oxidation of biomolecules and disruption effects [14]. Because radiation heat transfer from the light source to microbial cells and the surrounding environment may occur, the latest research has emphasized the additive effect of photochemical and photothermal reactions [15].

Pesto sauce can be contaminated by microorganisms arising from raw materials and manufacturing processes, with *Listeria monocytogenes* being a potential microbiological hazard [16]. Ready-to-eat (RTE) dips, sauces, and spreads have been involved in foodborne outbreaks, with *L. monocytogenes* as the causative agent and reason for recalls [17]. This pathogen deserves particular attention due to its ubiquitous nature and ability to survive and even grow in harsh environmental conditions, including in low water activity, a broad range of pH and temperatures (including under refrigeration), and the presence of preservatives [18]. A recent study by Salazar et al. [17] revealed that high-pressure processing (HPP) may be useful for reducing *L. monocytogenes* numbers in certain RTE products such as hummus, guacamole, and baba ghanoush. However, it was not effective in pesto and tahini.

Blue LEDs have demonstrated antibacterial activity against various pathogenic bacteria in laboratory media [19]. The effectiveness of blue light depends on the bacterial species, with *L. monocytogenes* being more susceptible than Gram-negative pathogens such as *E. coli*, *Salmonella enteritidis*, and *Shigella sonnei* [20]. The inactivation rates of *L. monocytogenes* vary depending on the illumination conditions. For example, Endarko et al. [20] reported a 1.5 log reduction at 123 J/cm^2^, which is equivalent to 4 h of exposure, while Kim et al. [21] achieved a 2.1 log reduction at 486 J/cm^2^ (equivalent to 7.5 h). Moreover, Olszewska et al. [22] reported a 2.8 log reduction at 2,672 J/cm^2^ (equivalent to 16 h). There are fewer studies on the effect of blue LEDs on the inactivation of this pathogen in foods. Kim et al. [23] recorded an approximately 1 log reduction in *L. monocytogenes* in fresh-cut mangos, similar to that reported by Sommers et al. [24] for chicken purge. Interestingly, Hyun and Lee [25] observed a reduction of 1.95 log CFU/g of *L. monocytogenes* on sliced cheese after 48 h, thus requiring long exposure times. There is currently no information on how *L. monocytogenes* responds to blue light in RTE sauces.

The efficacy of blue light relies on visible light in the 400–470 nm range as well as on the presence of photosensitizers and oxygen to generate ROS. One effective photosensitizer is curcumin, more generally known for its anti-inflammatory, antioxidative, antimicrobial, and anticancer properties [13]. Upon photosensitization, curcumin demonstrated antimicrobial potential against foodborne pathogens when applied to fresh-cut fruits [26,27], produce wash water or the surface of leafy vegetables [28]. This polyphenolic compound is found in the rhizome of *Curcuma longa* L., commonly known as turmeric. Turmeric is frequently used as a food additive [29,30]. When added to food, it could act as a significant source of photosensitizers while also enhancing biological activity [19]. However, further research is needed to validate these effects.

This study primarily aimed to assess the usefulness of blue light for the preservation of polyphenols, antioxidants, color, and pathogen (*L. monocytogenes*) reduction in a laboratory-prepared and turmeric-enriched kale pesto when applied alone or sequentially with sous-vide. Sous-vide is a thermal processing method that protects against the loss of moisture and the degradation of pigments, such as chlorophyll, in vegetables [31]. In this process, vegetables are vacuum-packed and thermally processed in sealed plastic bags, which minimizes nutrient loss [32]. A recent trend of applying low temperatures, e.g., from 40 to 70 °C in food processing [11], prompted us to use a low-temperature sous-vide to confer “fresh-like” characteristics to the pesto. However, some bacteria survive mild heat treatments, and *L. monocytogenes* is more heat resistant than most other non-spore-forming foodborne pathogens [33]. There is a lack of data on the thermal inactivation of pathogens in low-temperature conditions under a vacuum [11]. Thus, the findings of this study will be valuable for determining whether blue light and sous-vide treatments are suitable for reducing the amount of *L. monocytogenes* in pesto.

## 2. Results

### 2.1. Fate of Phenolic Compounds and Antioxidants Following Blue Light and Sous-Vide

The results concerning the TPC and antioxidative potential of both versions of the pesto following blue light illumination or processing using sous-vide as opposed to the sequential application of these two hurdles are presented in Table 1.

In freshly prepared pesto, the TPC ranged from 108 to 171 mg/100 g GAE dry matter (d.m.) when turmeric was not included, and from 150 to 208 mg/100 g GAE d.m. with the additive. In both cases, the lowest TP values were noted after the sous-vide, while the highest TPC was observed in the blue light–sous-vide samples (*p* < 0.05). Notably, blue light illumination improved the TPC by 20 to 30%. During storage, turmeric helped to retain relatively higher TP values, especially after blue-light treatment (*p* < 0.05). In turn, the blue-light-induced TP enhancement in fresh pesto was not preserved long-term when turmeric was absent. Contents of between 91 and 57 mg/100 g GAE d.m. were detected, with an activity loss of polyphenols of almost 40% in sous-vide samples. This is substantial compared with fresh pesto, which showed only an 8% loss as a result of the sous-vide treatment.

In evaluating the antioxidative capacity, the photochemiluminescence (PCL) method was used to measure both water-soluble (ACW) and lipid-soluble (ACL) components in the pesto samples (Table 1). In the water-soluble fraction, antioxidants such as flavonoids, ascorbic acid, and amino acids were detected. In contrast, the lipid-soluble fraction included tocopherols, tocotrienols, carotenoids, etc. The antioxidative activity of fresh raw pesto was estimated at 54.07 µmol TE/g dry matter (d.m.) when turmeric was included, compared with just 7.46 µmol TE/g d.m. without the additive (*p* < 0.05). The observed antioxidative activities were largely attributed to hydrophobic antioxidants, which were measured at 5.70 µmol TE/g d.m. without turmeric and 35.43 µmol TE/g d.m. with the additive.

Without turmeric, both sous-vide and blue light significantly increased the antioxidative capacity of pesto (*p* < 0.05). Notably, the combination of blue light and sous-vide resulted in the highest activity level, measuring 20.67 µmol TE/g d.m. (*p* < 0.05). However, when turmeric was added, sous-vide caused a decrease in activity levels of 12% in fresh pesto and 45% during storage, which were largely associated with the loss of hydrophilic antioxidants (*p* < 0.05). Because the blue light led to the increased concentration of antioxidants above 70 µmol TE/g d.m. (*p* < 0.05), it thus effectively compensated for the losses caused by sous-vide. In kale pesto, there was a correlation between the antioxidative activity levels obtained by the ACW, ACL, and PCL and the TPC (r = 0.707; r = 0.683; r = 0.706), indicating that the TP is slightly better correlated with water-soluble antioxidants.

### 2.2. Color Change Following Blue Light and Sous-Vide

Sous-vide led to an increase in kale pesto lightness (*L**), which was significant for both versions of the pesto after refrigerated storage (*p* < 0.05) (Table 2). Upon storing, turmeric-enriched pesto had a higher *L** than the corresponding samples that did not have this addition, with sous-vide pesto showing the highest *L** (*p* < 0.05). In turn, all blue-light-illuminated pesto had the lowest *L** (*p* < 0.05). However, *L** values increased when blue light and sous-vide were applied sequentially. This was significant for turmeric-enriched pesto following storage (*p* < 0.05). The color was further monitored using the color saturation (*C**) and hue angle (*h*°). The color of the pesto treated with blue light alone or with blue light and sous-vide was significantly less saturated than that of raw and sous-vide pesto (*p* < 0.05). The raw pesto had also the highest *h*°, resulting in a more intense hue of green. In contrast, the lowest *h*° was eventually observed for the samples treated with both blue light and sous-vide. On day 0, the total color difference (Δ*E*) in kale pesto ranged between 5 and 9 (*p* > 0.05). On day 14, the Δ*E* was found to be within the range of 6 to 17 (*p* < 0.05). Figure 1 depicts a color model in which red, green, and blue (RGB) were added to reproduce the colors of the pesto, and Appendix A shows the mean RGB values for each sample.

This model reveals that blue light had a reducing impact on the green coordinate; thus, this pesto was perceived as darker. Turmeric reduced the blue coordinate, keeping the first two relatively high throughout the storage, while all the RGB values in raw pesto decreased. Sous-vide, when also preceded by blue light, produced lighter-colored pesto, particularly for the turmeric-enriched samples. Because these color changes may be associated with water loss during processing, we also examined how the treatment conditions or the addition of turmeric affected the moisture content of kale pesto (Appendix A). Eventually, adding turmeric decreased the moisture content by 2%, whereas blue light either alone or sequentially with sous-vide reduced the contents by 6% and 8% in pesto with and without turmeric, respectively.

### 2.3. Survival of L. monocytogenes Following Blue Light and Sous-Vide

This part of the study assessed the survival of *L. monocytogenes* in both versions of pesto following each processing step (Figure 2). Sous-vide resulted in only a 0.6 log cell reduction in plain pesto. When turmeric was added, the reduction increased to 1.8 log CFU/mL, indicating a highly antimicrobial effect (Figure 2a). On day 0, we did not record cell reductions in the blue-light-illuminated pesto. In fact, prolonged illumination and, possibly, the accessibility of nutrients resulted in increased cell counts, although these were non-significant compared with the raw pesto (*p* > 0.05). The sequential application of blue light and sous-vide reduced *L. monocytogenes* levels by 1.7 logs. The presence of turmeric in the pesto further increased the efficacy of the treatment by an additional 2.0 logs, decreasing the cell counts from 6.0 to 2.3 log CFU/g (*p* < 0.05). Subsequently, all the cell counts declined, indicating that refrigerated storage time reinforced the observed effects, particularly for turmeric activated by blue light (Figure 2b).

### 2.4. Principal Component Analysis (PCA)

In component p1, the kale pesto with turmeric, PCL, and TPC clustered together, whereas the pesto without turmeric had an opposite loading (Figure 3). TPC was plotted between the PCL and blue light, implying their correlation. In component p2, blue light and *L. monocytogenes* had opposite loadings. *L. monocytogenes* and the sequential blue light and sous-vide treatment had similar loadings in p2, indicating a significant correlation. Turmeric was plotted in the same quadrant, suggesting that the presence of turmeric is necessary for the killing of *L. monocytogenes*. Sous-vide corresponded with lightness (*L**). The raw pesto seemed to correlate with the hue angle but had the least influence on the PCA model.

## 3. Discussion

### 3.1. Phenolic Compounds and Antioxidants of Processed Kale Pesto

The total phenolic content (TPC) and antioxidative activity have become increasingly important in assessing the reactive values of vegetables [5]. Thus, the current study included both measurements in the blue light and sous-vide-processed pesto as well as an analysis on how the addition of turmeric or storage affects its antioxidative power. Bioactive compounds might be susceptible to thermally induced oxidative degradation or thermally induced hydrolysis [34]. For instance, Murador et al. [35] reported the TPC in raw kale at the level of 49.20 ± 0.20 mg GAE/100 g; boiling decreased this significantly, by 31%, which was explained by leaching effects and the loss of activity of bioactive compounds. Similar results of reduced TPC and antioxidative activity levels of thermally processed kale were noted in other studies [36,37]. Interestingly, Korus [38] reported losses of 32% for polyphenols in kale after water blanching at 96–98 °C, which was introduced before drying as an approach to protect the kale’s bioactive compounds. By contrast, sous-vide tended to preserve the phenolic content and antioxidative activity to a greater extent than, for instance, boiling in various vegetables such as artichokes, green beans, broccoli, and carrots [31]. The current study revealed that during sous-vide, there was a TPC loss of less than 10% in fresh pesto and a significant reduction (of 40%) following refrigeration. Although small decreases in TPC are consistent with previous research on the sous-vide processing of green vegetables [31,39], they did not address the fate of phenolic compounds during storage. Nevertheless, these reductions could be attributed to the high water solubility of phenolic compounds, their thermal instability, and negative vacuum pressures [11].

Moreover, each processing step notably enhanced the antioxidative capacity of pesto, being highest after a combined treatment. An increase in the antioxidative activity of processed vegetables may be attributed to the enhanced capabilities of naturally occurring antioxidants or the formation of novel compounds with antioxidative activity during a Maillard reaction [10]. Moreover, this effect could also be due to the breakdown of compounds, leading to a variation in phenolic content and composition or the possible interaction of components contributing to the antioxidative activity [35]. As for sous-vide, some bioactive compounds may be better preserved in vegetable products due to their limiting oxidation under vacuum conditions [11]. Moreover, Nartea et al. [40] recently demonstrated that boiling, steaming, or sous-vide enhanced the release of antioxidants, such as carotenoids and tocopherols, from cauliflower. They found that longer processing times increased the extractability of these compounds and attributed this effect to the softening of the vegetable’s matrix. The bioactivity of kale pesto was not only dependent upon the processing step but also on the addition of turmeric. Although there was significant bioactivity enhancement by turmeric, these compounds were labile during sous-vide and decreased by 12% in fresh pesto and by 45% during refrigerated storage. This was particularly evident for hydrophilic antioxidants, confirming their instability under heat and vacuum, followed by refrigeration.

Given the complexity of factors contributing to the antioxidative power of processed vegetables, blue light appears to have a positive impact on the antioxidative activity and the phenolic compounds present in kale pesto, including those coming from turmeric addition. A possible explanation could be among the abovementioned, including the increased retention and release of antioxidants due to a mild and timely illumination process. Importantly, blue light effectively compensated for the losses caused by sous-vide while retaining these antioxidants during storage. Blue light as a (pre)treatment for preserving the bioactive components of vegetables has not been thoroughly studied. However, the impact of blue LEDs on the quality of some fruits has been documented. For instance, Kim et al. [41] revealed that the total flavonoid content in illuminated fresh-cut papayas was 1.5–1.9 times higher than that of non-illuminated fruits. The tissue cells received 0.9–1.70 kJ/cm^2^, corresponding to 24–48 h of illumination. It was explained that this effect is probably due to the stimulation of light on the production of primary and secondary metabolites, which are implicated in the defense against ROS created during blue-light illumination. Another study determined a positive effect of blue light at an intensity of 40 µmol m^−2^ s^−1^ on entire bayberries, leading to the accumulation of anthocyanin after 8 days compared with non-illuminated fruits [42]. In contrast, Liu et al. [43] reported that orange juice treated with brief exposure to blue light showed reduced total phenolic and flavonoid contents, along with a decline in antioxidative activity. This was attributed to the post-treatment loss of water-soluble antioxidants (such as ascorbic acid) generating oxidizing ROS. These findings demonstrate that the effect of blue light varies depending on the matrix and illumination conditions, and the precise mechanisms driving the biological activity remain to be identified.

### 3.2. Color of Processed Kale Pesto

Color is one of the most important parameters for evaluating food quality in light-based technologies [44]. In fact, both thermal- and oxidizing-ion-based treatments may negatively impact the color [10]; likewise, the use of curcumin as an additive may lead to color deterioration [19]. The ability of blue light to change the color of foods has been established. For instance, Kim et al. [41] found that fresh-cut papayas exposed to blue light had the lowest *L***a***b** values, resulting in a higher Δ*E* compared with non-illuminated fruits. The highest Δ*E* was close to 37 after receiving 1.30 kJ/cm^2^, corresponding to 36 h of illumination. Notably, color changes were also observed in non-illuminated papayas during storage for the same duration as the illuminated fruits (24 to 48 h), suggesting that cut processing and ripening largely contributed to these alterations. Likewise, Chai et al. [45] studied the effect of blue-light treatment on the quality of fresh-cut pears and found that it resulted in a lower *L** and higher yellowness. Unfortunately, during the 6-day storage, the *L**, *b**, and hardness of the pears deteriorated. Ghate et al. [44] reported significant decreases in the yellowness of fresh-cut pineapples due to blue LEDs at 7950 J/cm^2^ (8.66 h). It was explained that the color changes were due to the absorption of light by the pigment in the pineapple β-carotene and its photodegradation to β-carotene radicals or free-radical adducts. The breakdown of pigments, mainly chlorophylls, in kale most likely contributed to the color shift due to blue-light illumination, which was accompanied by moisture loss. During processing, chlorophyll can convert to olive-green pheophytins and pyropheophytins, and when the cell membrane of the tissues is disrupted, enzyme activity further contributes to these changes [39]. Despite this, the formation of Maillard-derived melanoidins may also be responsible for the color alterations frequently reported in infrared-treated fruits and vegetables, which are associated with higher antioxidative activity [10]. Nevertheless, the right optimization of the blue-light treatment could potentially minimize these color alterations. Interestingly, the color changes caused by the processing of kale pesto were lower than those of some cooked green vegetables, including sous-vide treatment at above 80 °C or by steaming [39]. It was also observed that the addition of turmeric had less of an impact on the color change but rather helped to maintain the initial color throughout the storage period.

### 3.3. Safety of Processed Kale Pesto

Despite the potential of non-thermal technologies, developing microbiologically safe pesto sauce with blue LEDs is challenging. There are limitations to using this technology in food products. These limitations include the low penetration associated with visible light, as well as the absence of enzyme-inhibitory ability in foods [44]. Bacterial cells may show reactivation potential following non-thermal and antimicrobial treatments [46,47]. In particular, sublethal stress factors may result in variable susceptibility, and these aspects require further investigation. Combining non-thermal treatments with other food preservation methods could help overcome these drawbacks. According to Hyun and Lee [19], the combined effects of LEDs with temperature, antimicrobials, and photosensitizers are considered the most promising. In the case of *Listeriae*, combining pulsed light with other mild treatments, such as pulsed electric fields and thermosonication, has been shown to improve treatment efficiency against *Listeria innocua* [48]. The current study found that blue light under the conditions provided was insufficient to inactivate *L. monocytogenes* in kale pesto. A crucial factor for the inactivation was the addition of turmeric. As a natural source of photosensitizers, it was highly antibacterial when photo- and thermally activated with blue light and sous-vide.

The potent compound of turmeric, curcumin (or curcumin-rich extracts), has been revealed as an efficient photosensitizer for *L. monocytogenes* in solutions like PBS, resulting in >6.0 log reductions in cell numbers [49,50]. Curcumin’s lipophilic structure allows it to be directly inserted into the lipid bilayer, increasing its permeability [51]. Enhanced permeability of bacterial cell membranes leads to integrity disruption and, eventually, cell death. ROS implicated in the photosensitization of curcumin are anticipated to be formed in higher quantities and to act more efficiently in exerting antimicrobial effects [14]. Thus, it is reasonable to expect this photo-antimicrobial effect in different foods. For instance, photosensitization with curcumin has been demonstrated on the surface of fresh-cut pears, resulting in a 3.43 log CFU/g reduction in *L. monocytogenes* within 10 min [45]. Gao and Matthews [52] reported that 32.1 kJ/m^2^ of illumination with 300 ppm of curcumin (30 × MIC) resulted in a 2.9 log CFU/cm^2^ reduction in *L. monocytogenes* on chicken skin. Finally, antibacterial tests in wells of a 48-well plate with orange juice demonstrated that a low dose of 1.5 J/cm^2^ and the supramolecular inclusion complex of curcumin and β-cyclodextrin achieved a 2.9 log reduction [43]. Although these tests used small food slices or volumes and were primarily aimed at surface decontamination, they demonstrated that curcumin, a food-compatible photosensitizer, can improve the efficacy of blue LEDs. Further research, for example, on their operating conditions, is needed to improve the interaction with food samples, especially with opaque liquids and high-volume foods, as in the current study with pesto sauce. In addition, to make photodynamic-based procedures more effective, the barrier of the limited diffusion of photosensitizers and cytotoxic oxygen species must be overcome. A previous study by Gazmeh et al. [53] revealed that the diffusion coefficients of curcumin increase with increasing temperature due to the increased tendency of water molecules to surround the curcumin molecule. Hence, we hypothesize that the increase in temperature as a result of sous-vide increased the dispersion of curcumin, greatly improving photodynamic efficiency by allowing more molecules to reach and act on cells.

### 3.4. PCA

The PCA model indicated that the reduction in *L. monocytogenes* in pesto is influenced by variables such as blue light, sous-vide, and the addition of turmeric, implying the photothermal activation of its active compound(s). Turmeric was also associated with antioxidants and phenolic compounds present in the kale pesto. The obtained results thus suggest that the presence of a variety of bioactive compounds in turmeric contributes to the observed effects in two ways: first, as photosensitizing agents, and second, as antioxidants. This is in line with Bonifácio et al. [14], who emphasized the importance of other curcuminoids in *C. longa*, such as those in the structure of bisdemethoxycurcumin and demethoxycurcumin, which may act as antioxidants and exert protective effects against oxidation induced by curcumin itself.

## 4. Materials and Methods

### 4.1. Preparation of Kale Pesto and Processing

Kale (*Brassica oleracea* L. var. *sabellica* L.) was obtained from a local discount store in Olsztyn, Poland, as was olive oil, salt, and turmeric rhizome. The kale was cleaned by removing the wooden stems, washing the leaves in tap water, and gently shaking them in sieves until dry. Kale leaves were ground in a Robot Coupe^®^ (R 301 ultra D; Vincennes, France) for 30 s. The rhizome was washed, peeled, grated, and homogenized at 2000 rpm for 1 min (HO 4A, Bühler; GmbH, Hechingen, Germany). The turmeric homogenate was then added (~8% *w*/*w*), along with olive oil (10% *w*/*w*) and salt (2% *w*/*w*). All the ingredients were blended using a Kenwood planetary robot (KM080; Havant, UK) for 15 s. The plain pesto (without turmeric) included only kale leaves, olive oil, and salt. Pesto that avoided processing was divided into portions of 25 g and vacuum-packed in 75 μm thick poly-amide/polyethylene (PA/PE) pouches with dimensions 150 × 200 mm (Hendi BV, Rhenen, The Netherlands) using a chamber vacuum sealer (Edesa VAC-20 DT; Barcelona, Spain). For the sous-vide, 25 g of pesto was vacuum-packed and processed using a water bath with an immersion circulator and a temperature sensor (Diamond Z; Julabo GmbH, Seelbach, Germany). For the blue light, the prepared pesto was spread flat in a stainless steel container positioned 100 mm away right beneath the light source and illuminated for 12 h at 65 mW/cm*^2^* using 18 × 1 W LED bulbs (8 × 460 nm/9 × 430 nm/1 × 410 nm; Epistar Corp.; Hsinchu, Taiwan) with an aluminum heat sink (Figure 4). When applying the two treatments sequentially, the pesto was first exposed to blue light, followed by vacuum-sealing the 25 g portions and cooking them sous-vide. All samples were cooled in an ice-water bath at 0 °C for 10 min before analysis or stored at 4 °C for 14 days. The analyses were performed on day 0 and after 14 days of storage at 4 °C.

### 4.2. Extract Preparation

One mL of 80% methanol was introduced to 100 mg of the sample. Subsequently, the solution underwent extraction using an ultrasound-assisted method [54]. Each solution underwent 0.5 min of vortexing, followed by 0.5 min of sonication and 5 min of centrifugation at 5000× *g* and 4 °C. This process was iterated 5 times, with the residue being re-suspended in 1 mL of fresh 80% methanol. The resultant supernatants were combined, and the extracts (approx. concentration 20 mg/mL) were stored at −80 °C before analyses of the total phenolic content (TPC) and antioxidative activity.

### 4.3. Total Phenolic Content (TPC)

Assessment of the total phenolic content (TPC) was carried out in a microplate reader (Infinite M1000; Tecan, Männedorf, Switzerland) [54]. For this, 0.25 mL of the extract, at a concentration of 20 mg/mL, was combined with 0.25 mL of Folin’s phenol reagent (CAS no. 47641; Sigma-Aldrich, St. Louis, MO, USA), 0.5 mL of saturated sodium carbonate (Na_2_CO_3_; CAS no. 144-55-8; Sigma-Aldrich), and 4 mL of water. The resultant mixture underwent a 25 min incubation at room temperature (21 °C) followed by centrifugation at 2000× *g* for 10 min. Subsequently, the absorbance of the supernatant was determined at 725 nm. A calibration curve was established with a gallic acid (CAS no. 149-91-7, Sigma-Aldrich) standard within a concentration range of 0.01–0.70 mg/mL (y = 1.7499x + 0.0321; R^2^ = 0.999). As a result, the TPC has been expressed as mg gallic acid equivalents per 100 g of dry matter (d.m.).

### 4.4. Antioxidative Activity

Antioxidative activity was assessed using the photochemiluminescence (PCL) method with the PHOTOCHEM^®^ system from Analytik Jena (GmbH+Co. KG, Jena, Germany) [55]. The Water-soluble (ACW) and Lipid-soluble (ACL) Antioxidant Capacity protocols were utilized to determine the extracts’ scavenging activity against superoxide anion radicals (O_2_^•−^). The luminal reagent, Trolox stock, and working solutions were prepared following the manufacturer’s instructions. A stock solution of luminal reagent was prepared by adding 750 µL of buffer (as indicated in kit number 2) and vortexing for 0.5 min. In contrast, the Trolox stock solution was diluted with 490 µL of buffer (from kit number 1). Following this, 10 µL of the Trolox solution was further diluted with 990 µL of buffer number 1, creating a 1:100 dilution. Each 10 µL of this working solution contains 1 nmol of calibration Trolox. The extracts were added at concentrations that resulted in the luminescence falling within the range limits of the standard curve. Trolox ((±)-6-hydroxy-2,5,7,8-tetramethylchromane-2-carboxylic acid; CAS no. 53188-07-1; Sigma-Aldrich) served as a standard within the 0.25–1.00 nmol range (R^2^ = 0.9991 in ACW; R^2^ = 0.9956 in ACL). To validate the measurement, two blanks and a set of diluted Trolox samples were typically run as per the PCL protocol recommendations. The results were expressed as µmol Trolox equivalents (TE)/100 g d.m.

### 4.5. Color

The color was evaluated with a Minolta CR-400 Chroma Meter (Konica Minolta Sensing Inc., Osaka, Japan), using a standard 2° observer and a D65 illuminator. The instrument was calibrated with the use of a white ceramic plate provided by the manufacturer. The measurements were expressed using the CIELAB (*L***a***b**) scale, in which color coordinates refer to the lightness (*L**), red-green coordinate (*a**), and yellow-blue coordinate (*b**). The color saturation change Chroma (*C**), hue angle (*h*°), and total color difference (Δ*E*) were calculated as follows:
(1)$$C*=\sqrt{{a*}^{2}+{b*}^{2}}$$
(2)h°=((ATAN(b*/a*)×360°/2×3.14))+180°
(3)$$\Delta E*=\sqrt{{\Delta L*}^{2}+{\Delta a*}^{2}+{\Delta b*}^{2}}$$
where:

Δ*L** = the lightness difference

Δ*a** = the redness difference

Δ*b** = the yellowness difference

The CIELAB color space was converted into an RGB color model, in which red, green, and blue were added to reproduce the colors of the kale pesto using the Color Designer tool (https://colordesigner.io, accessed on 21 October 2024).

### 4.6. L. monocytogenes Strains and Pesto Inoculation

The *L. monocytogenes* strains used in this study were ATCC 15313, ATCC 19112, and ATCC 19115. All the strains were stored at −80 °C in cryovials with 30% (*v*/*v*) glycerol, and they were streaked out and cultured on tryptic soy agar plates (TSA; Merck, Darmstadt, Germany) before experiments. A cocktail was prepared using serotypes 1/2a, 1/2c, and 4b. This selection was made to address the variations in virulence and stress resistance, including to heat [33,56]. Notably, these serotypes were identified in both food and clinical isolates, with serotype 4b being responsible for the majority of human listeriosis cases [33]. For experiments, single colonies of each strain were picked and cultured individually in tryptic soy broth (TSB; Merck KGaA, Darmstadt, Germany) overnight at 37 °C. The overnight cultures were centrifuged (5,000× *g*, 3 min). Then, the harvested cells were washed with phosphate buffer saline (PBS, Sigma-Aldrich) and resuspended in the same solution to a volume of 1 mL, which was maintained at an optical density at 600 of 0.2 (×10^8^ CFU/mL) using a microplate reader (Multiskan™ GO; Thermo Fisher, Waltham, MA, USA). The suspensions were serially diluted in sterile saline (0.85% NaCl) and plated onto TSA to verify the cell numbers. Then, equal aliquots of the resulting suspensions were combined to form a cocktail. The prepared *L. monocytogenes* cocktail was added to the pesto at a ratio of 1:100, achieving a final concentration of 6 log CFU/g. The pesto was hand- stirred for 2 min to homogenize it, and it was then subjected to blue light in a separate container. PA/PE pouches were filled with 25 g of inoculated pesto and the top of the pouch was vacuum sealed using a Silvercrest^®^ SV125 mini sealer (Hoyer Handel GmbH, Hamburg, Germany) in a class II biosafety cabinet. Thus, separate pesto pouches were prepared and subjected to processing in a dedicated sous-vide device (Hendi GN 2/3, Rhenen, The Netherlands) to uphold safety standards.

### 4.7. Enumeration of L. monocytogenes

Pouches with inoculated pesto were cut open and 10 g aliquots of each pesto sample were placed into 720 mL stomacher bags. The samples were homogenized for 1 min with 90 mL of sterile saline, serially diluted in the same diluent, and plated onto Oxford agar with Listeria Selective Supplement (Merck) containing acriflavin, cefotetan, colistin sulfate, cycloheximide, and fosfomycin. Before colony counting, the plates were incubated at 37 °C for 24 h. The lowest level of enumeration was 2 log CFU/g.

### 4.8. Statistical Analysis

Data analysis was conducted using TIBCO^®^ Statistica™ ver. 13.3 (TIBCO Software Inc., Tulsa, OK). One-way ANOVA, followed by Tukey’s post hoc test was established to test for differences between the processing steps and the pesto version (with and without turmeric) at a significance level of *p* < 0.05. Principal component analysis (PCA) was conducted to show the amount of clustering among tested variables that was associated with the antioxidative and anti-listerial activities and the color of the kale pesto. The Pearson correlation coefficient (r) was used to evaluate the direction and strength of the tested variables. Hierarchical cluster analysis was performed on the variables to verify their qualification for the PCA.

## 5. Conclusions

The current study is the first attempt to evaluate the impact of blue light and sous-vide on the quality and safety of kale pesto with turmeric as a bioactive ingredient. On one hand, turmeric acts as a bioactivity enhancer for pesto, while on the other hand, it is as a source of photosensitizing molecules that are highly antibacterial after photothermal activation. Blue light helped to retain the antioxidative capacity of turmeric-enriched pesto, compensating for losses caused by the sous-vide treatment. However, it exerted a more pronounced effect on the color of pesto, which warrants further investigation into how this translates to the sensory attributes and consumer acceptability of such products. Although blue light has only recently entered the research and development stage, this study suggests that it could be an effective (pre)treatment for RTE products like pesto if the limiting processes and product factors are considered. Importantly, this study provides novel information on the reduction in *L. monocytogenes* in RTE pesto sauces and can be used as a starting point to assess the use of blue light and sous-vide in other RTE products.

## Figures and Tables

**Figure 1 molecules-29-05831-f001:**
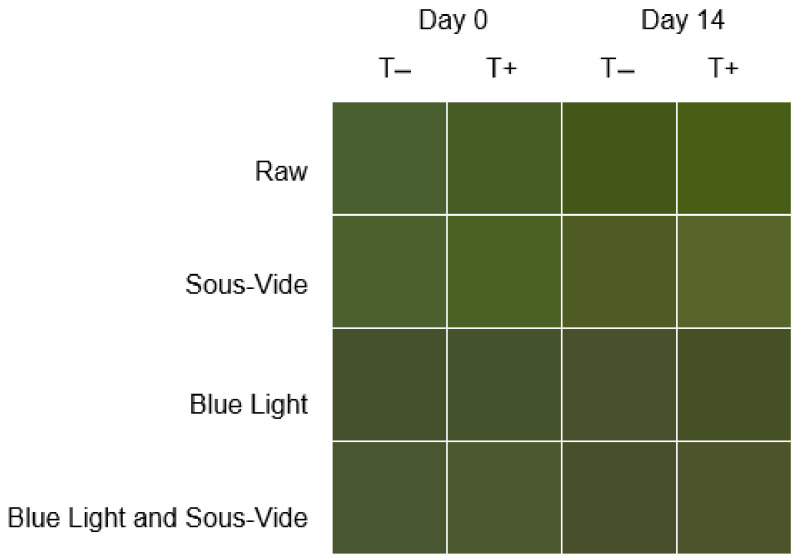
Color model in which red, green, and blue were added to reproduce the colors of processed kale pesto with (T+) and without (T−) turmeric.

**Figure 2 molecules-29-05831-f002:**
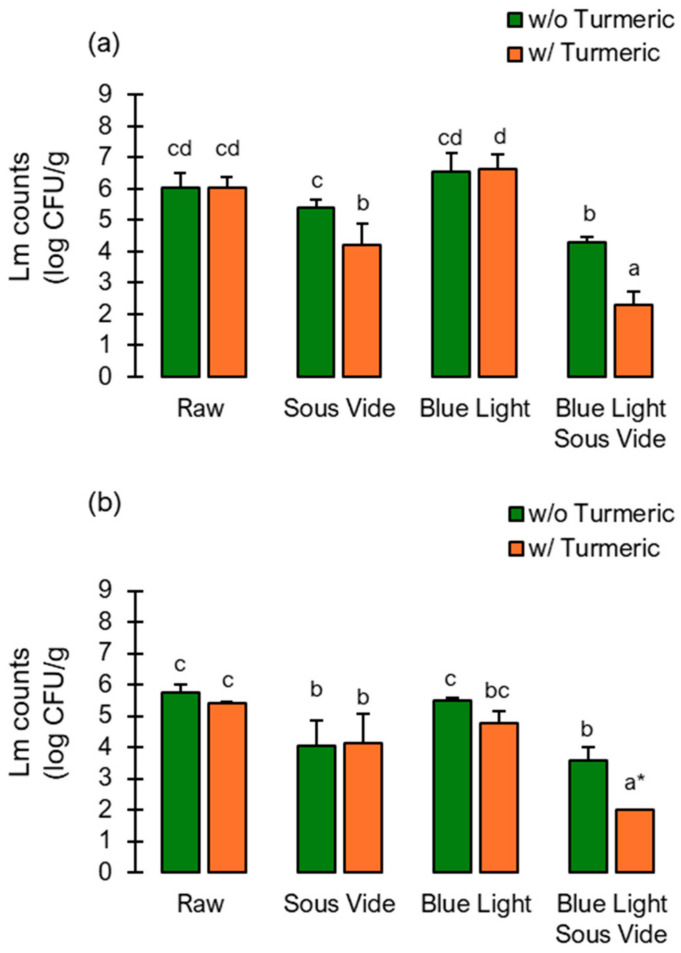
Cell counts of *L. monocytogenes* (ATCC 15313, ATCC 19112, ATCC 19115) in kale pesto with and without the addition of turmeric were measured after each processing step at day 0 (**a**) and day 14 of storage under refrigeration (4 °C). (**b**) These included sous-vide at 60 °C for 8 min, blue light for 12 h at 65 mW/cm^2^ from 410–460 nm, and a sequential application of blue light and sous-vide. Raw pesto avoided processing and was vacuum-packed only. * indicates that most of the cell counts determined were below the detection limit. Values are expressed as means (n = 3) ± standard deviations. Mean values with different letters (^a–d^) are statistically different (*p*-value < 0.05). * indicates detection level.

**Figure 3 molecules-29-05831-f003:**
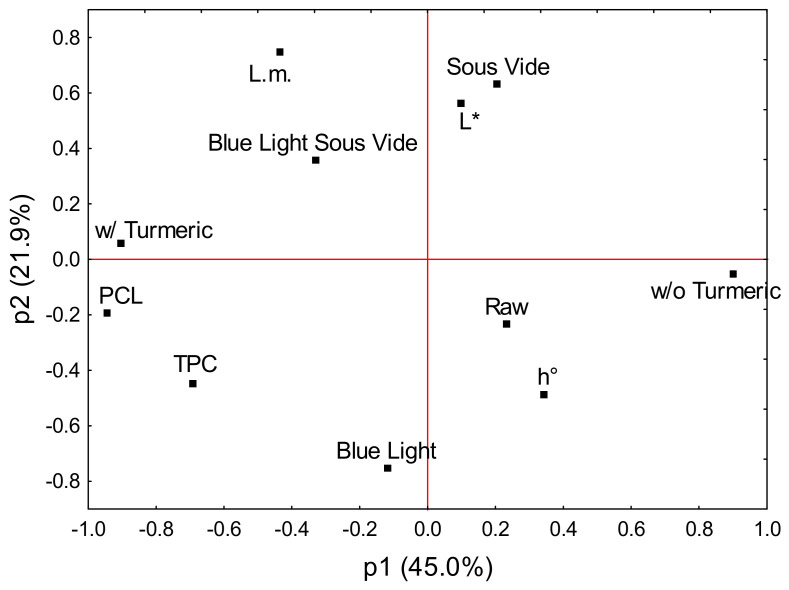
The loading scatterplot (p1 vs. p2) showing clustering among the variables qualified based on the hierarchical cluster analysis. Variables placed close to each other influence the PCA model in a similar way, which also indicates that they are correlated. The further away a variable from the origin, the more influential the variable is in determining the PCA model. The following continuous variables were analyzed: TPC, PCL, lightness (*L**), hue angle (*h*°), and *L. monocytogenes* (L.m.), indicating the pathogen responsiveness. The kale pesto version and processing steps were selected as categorical variables.

**Figure 4 molecules-29-05831-f004:**
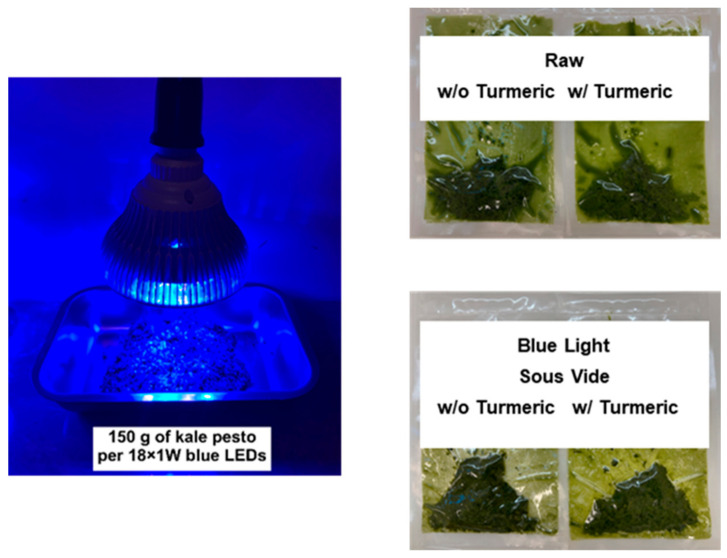
Blue LEDs assembled over the kale pesto and vacuum-sealed pesto products following blue light and sous-vide.

**Table 1 molecules-29-05831-t001:** Total phenolic content and antioxidative activity of processed kale pesto with and without the addition of turmeric.

Storage	Pesto Version	Processing Step	TPC (mg GAE/100 g d.m.)	ACW TE (µmol/g d.m.)	ACL TE (µmol/g d.m.)	PCL TE (µmol/g d.m.)
Day 0	w/o Turmeric	Raw	117.60 ± 3.48 ^ab^	1.76 ± 0.03 ^a^	5.70 ± 0.17 ^a^	7.46 ± 0.20 ^a^
		Sous-vide	107.60 ± 5.08 ^a^	4.13 ± 0.14 ^b^	12.76 ± 0.22 ^b^	16.90 ± 0.33 ^b^
		Blue Light	146.89 ± 9.76 ^bc^	3.48 ± 0.02 ^b^	11.53 ± 0.39 ^b^	15.01 ± 0.40 ^b^
		Blue Light and Sous-vide	171.06 ± 3.70 ^c^	6.08 ± 0.07 ^c^	14.59 ± 0.05 ^b^	20.67 ± 0.03 ^c^
	w/Turmeric	Raw	159.07 ± 20.05 ^c^	18.65 ± 1.11 ^e^	35.43 ± 1.85 ^c^	54.07 ± 2.47 ^e^
		Sous-vide	149.80 ± 15.95 ^bc^	12.60 ± 0.40 ^d^	35.01 ± 1.62 ^c^	47.61 ± 2.01 ^d^
		Blue Light	205.37 ± 10.50 ^d^	32.10 ± 0.13 ^f^	40.71 ± 1.58 ^d^	72.82 ± 1.59 ^f^
		Blue Light and Sous-vide	207.90 ± 11.66 ^d^	19.76 ± 0.57 ^e^	36.68 ± 0.85 ^c^	56.45 ± 0.45 ^e^
Day 14	w/o Turmeric	Raw	91.04 ± 6.41 ^bc^	1.59 ± 0.06 ^a^	5.25 ± 0.07 ^a^	6.84 ± 0.13 ^a^
		Sous-vide	56.59 ± 2.97 ^a^	3.48 ± 0.14 ^b^	11.49 ± 0.35 ^b^	14.97 ± 0.49 ^bc^
		Blue Light	71.81 ± 22.87 ^ab^	3.13 ± 0.11 ^b^	9.62 ± 0.28 ^b^	12.75 ± 0.38 ^b^
		Blue Light and Sous-vide	88.68 ± 4.15 ^bc^	5.14 ± 0.22 ^c^	12.65 ± 0.54 ^b^	17.79 ± 0.73 ^c^
	w/Turmeric	Raw	110.35 ± 2.39 ^cd^	18.12 ± 0.29 ^e^	32.75 ± 2.22 ^d^	50.88 ± 2.15 ^e^
		Sous-vide	85.47 ± 2.13 ^bc^	7.08 ± 0.26 ^d^	20.97 ± 0.76 ^c^	28.05 ± 0.84 ^d^
		Blue Light	175.12 ± 6.87 ^e^	31.70 ± 0.61 ^g^	40.07 ± 2.25 ^e^	71.77 ± 2.68 ^f^
		Blue Light and Sous-vide	129.80 ± 4.35 ^d^	19.65 ± 0.76 ^f^	34.70 ± 1.50 ^d^	54.34 ± 2.24 ^e^

GAE—gallic acid equivalents; d.m.—dry matter; TE—Trolox equivalents; PCL—sum of ACW and ACL. The antioxidative capacity of the water-soluble (ACW) and lipid-soluble (ACL) components. Values are expressed as means (n = 3) ± standard deviations. Mean values with different letters (^a–g^) in each columns for days 0 and 14 are statistically different (*p*-value < 0.05).

**Table 2 molecules-29-05831-t002:** Color measurements of the processed kale pesto with and without the addition of turmeric.

Storage	Pesto Version	Processing Step	*L**	*C**	*h*°	Δ*E*
Day 0	w/o Turmeric	Raw	37.35 ± 3.08 ^d^	29.79 ± 5.05 ^bc^	125.71 ± 1.47 ^e^	
		Sous-vide	38.03 ± 1.32 ^d^	31.71 ± 4.20 ^c^	123.86 ± 1.91 ^d^	4.86 ± 3.78 ^a^
		Blue Light	32.55 ± 1.55 ^a^	23.99 ± 1.67 ^a^	122.09 ± 1.29 ^bc^	8.98 ± 4.38 ^a^
		Blue Light and Sous-vide	34.48 ± 1.74 ^abc^	22.38 ± 1.07 ^a^	122.05 ± 0.59 ^bc^	9.05 ± 4.84 ^a^
	w/Turmeric	Raw	36.10 ± 1.63 ^cd^	33.45 ± 4.86 ^cd^	122.41 ± 1.13 ^cd^	
		Sous-vide	37.75 ± 1.00 ^d^	36.71 ± 4.33 ^d^	120.49 ± 1.26 ^ab^	7.15 ± 4.59 ^a^
		Blue Light	33.49 ± 1.18 ^ab^	25.12 ± 1.59 ^ab^	121.96 ± 0.91 ^bc^	8.89 ± 4.66 ^a^
		Blue Light and Sous-vide	35.75 ± 1.08 ^bcd^	25.27 ± 1.37 ^ab^	120.06 ± 0.80 ^a^	8.56 ± 5.00 ^a^
Day 14	w/o Turmeric	Raw	34.01 ± 0.33 ^b^	38.17 ± 1.13 ^d^	118.80 ± 0.34 ^d^	
		Sous-vide	36.48 ± 1.29 ^c^	33.60 ± 3.12 ^c^	116.37 ± 1.03 ^bc^	5.83 ± 3.07 ^a^
		Blue Light	32.67 ± 0.93 ^ab^	22.86 ± 1.70 ^a^	118.72 ± 0.81 ^d^	15.41 ± 1.63 ^b^
		Blue Light and Sous-vide	32.21 ± 1.59 ^a^	22.61 ± 2.19 ^a^	115.21 ± 1.46 ^b^	15.91 ± 2.46 ^b^
	w/Turmeric	Raw	36.79 ± 0.43 ^c^	41.58 ± 2.38 ^d^	117.78 ± 0.49 ^cd^	
		Sous-vide	40.00 ± 0.74 ^d^	34.60 ± 4.33 ^c^	116.24 ± 1.08 ^bc^	9.26 ± 3.33 ^a^
		Blue Light	32.26 ± 0.85 ^a^	27.55 ± 1.74 ^b^	116.64 ± 1.03 ^bc^	14.83 ± 2.57 ^b^
		Blue Light and Sous-vide	33.87 ± 1.49 ^b^	25.04 ± 1.90 ^ab^	113.02 ± 2.49 ^a^	17.22 ± 2.78 ^b^

Values are expressed as means (n = 10) ± standard deviations. Mean values with different letters (^a–e^) in each column for days 0 and 14 are statistically different (*p*-value< 0.05).

## Data Availability

The data that support the findings of this study are openly available in Zenodo at https://doi.org/10.5281/zenodo.14002040.

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
