# Peer review of "Improvement of Selected Quality and Safety Traits in Turmeric-Enriched Kale Pesto Using Blue Light and Sous-Vide"

_molecules, 2024, doi:10.3390/molecules29245831_

Round 1

Reviewer 1 Report

Comments and Suggestions for Authors

The article submitted by Olszewska et al. displays and discusses the results of an investigation of the effects of sous-vide and blue-light irradiation alone or sequentially combined upon the phenolic fraction and the colour attributes of a ready-to-eat (RTE) sauce (Kale pesto) with or without turmeric as an ingredient. The study also includes a series of assays assessing the effectiveness of the abovementioned treatments to lower the load of Listeria monocytogenes artificially inoculated into the product.

Despite the accuracy of the experimental results, I have two main concerns with regards to the soundness of the study itself and its applicability in an industrial context. Firstly, is the power penetration of blue light into the sauce specially if it is intended to be applied in an industrial context where large quantities are processed. The second one is the microbiological assays in relation to L. monocytogenes. As described, and considering the typology of matrix, none of the final products reaches acceptability regarding the current in-force food safety criteria for RTE of the Regulation (EC) 2073/2005 and, therefore, its application as an antimicrobial strategy for this product is rather limited. Besides, the EURL-Lm specifications of the technical document on challenge tests for L. monocytogenes should have been followed (ANSES & EURL-Lm, 2021). Additionally, the article lacks on discussion in some parts and contain unnecessary information in others.

Consequently, the manuscript cannot be published in its current form and must be thoroughly modified in contents and format.

TITLE

Considering the study, the tile is somehow misleading, specially with the term quality. Please modify it accordingly.

INTRODUCTION

L33-34 Not relevant for the purposes of the study. Delete.

L42-48 Reduce the lines related to Kale’s nutritional aspects.

L49-51 These are general methods used in food analyses, not just for pesto sauces.

L51 Delete repetition.

L83 ubiquitous nature

More information regarding previous studies involving pathogen inactivation using blue light diode-activated ROS in RTE products should be included.

RESULTS AND DISCUSSION

L120-121 “it is believed that”, is not scientifically appropriated. Please take special care with these writing.

Table 3 Is not a Table but a Figure and should be placed in Supplementary Information.

Up to this point of the manuscript, the study has been focused on organoleptic and nutritional components’ aspects of kale pesto after sous-vide and/or blue light irradiation. However, there is no sensory panel evaluation to assess changes and acceptability/rejection of the product after processing, and it would be desirable to include these assays if it is intended to put the product into the market.

L311-320 The hypothesis of VBNC cells are not sufficiently supported with the presented results. Please reconsider this fragment but I would recommend to delete it.

L321-329 This section has no discussion and must be added.

MATERIALS AND METHODS

L350 Description column in Table 4 should be placed in the main body of the manuscript.

L360 30 s vortex …. 5 min of centrifugation. Please uniform the same unit format.

L365-370 This section should be merged into the appropriate sections of Materials and Methods.

L363 How was this concentration measured?

L386 Give some details about the preparation of the working solutions.

L397 Give CIE full name.

L411-424 No information regarding the preparation of the bacterial stock cultures, working cultures, the method to adjust the optical density of the culture and the corresponding value in CFU/mL of the Abs600 = 0.8 AU.

Author Response

The article submitted by Olszewska et al. displays and discusses the results of an investigation of the effects of sous-vide and blue-light irradiation alone or sequentially combined upon the phenolic fraction and the colour attributes of a ready-to-eat (RTE) sauce (Kale pesto) with or without turmeric as an ingredient. The study also includes a series of assays assessing the effectiveness of the abovementioned treatments to lower the load of Listeria monocytogenes artificially inoculated into the product.

Despite the accuracy of the experimental results, I have two main concerns with regards to the soundness of the study itself and its applicability in an industrial context. Firstly, is the power penetration of blue light into the sauce specially if it is intended to be applied in an industrial context where large quantities are processed. The second one is the microbiological assays in relation to L. monocytogenes. As described, and considering the typology of matrix, none of the final products reaches acceptability regarding the current in-force food safety criteria for RTE of the Regulation (EC) 2073/2005 and, therefore, its application as an antimicrobial strategy for this product is rather limited. Besides, the EURL-Lm specifications of the technical document on challenge tests for L. monocytogenes should have been followed (ANSES & EURL-Lm, 2021). Additionally, the article lacks on discussion in some parts and contain unnecessary information in others. Consequently, the manuscript cannot be published in its current form and must be thoroughly modified in contents and format.

Response: Thank you for bringing these concerns to our attention. Recent blue light is in the research and development stage, thus although our study suggests it could be an effective (pre)treatment on RTE products, numerous limiting process and product factors should be considered that we have addressed in the revised manuscript. Regarding L. monocytogenes, published information is lacking pertaining to how L. monocytogenes responds to mild treatments imposed by blue light and sous-vide at low temperatures in RTE dips, sauces, and spreads. Hence, we tried to better explain the rationale behind the selection of this particular pathogen in the introduction (Lines 100-113). The study was designed to estimate whether those new treatments are appropriate for the reduction of the most heat-resistant vegetative pathogen (L. monocytogenes) and can be used as a starting point to assess the use of blue light and sous-vide in other RTE products (Conclusions).

TITLE

Considering the study, the tile is somehow misleading, especially with the term quality. Please modify it accordingly.

Response: Thank you for bringing this to our attention. The title has been modified to better reflect the purpose and outcome of the study to: Improvement of Selected Quality and Safety Traits in Turmeric-enriched Kale Pesto using Blue Light and Sous-Vide.

INTRODUCTION

L33-34 Not relevant for the purposes of the study. Delete.

Response: The sentence has been deleted.

L42-48 Reduce the lines related to Kale’s nutritional aspects.

Response: The lines related to kale’s nutritional aspects have been reduced. Line 41.

L49-51 These are general methods used in food analyses, not just for pesto sauces.

Response: This comment has been addressed, thank you. Line 45.

L51 Delete repetition.

Response: The repetition has been deleted, thank you.

L83 ubiquitous nature

Response: The omnipresence has been replaced by ubiquitous. Line 70.

More information regarding previous studies involving pathogen inactivation using blue light diode-activated ROS in RTE products should be included.

Response: More information regarding the blue light efficacy against the pathogen has been added. Lines 77-89.

RESULTS AND DISCUSSION

L120-121 “it is believed that”, is not scientifically appropriated. Please take special care with these writing.

Response: This concern has been addressed. The paragraph has been rewritten in accordance to the suggestions of other reviewers.

Table 3 Is not a Table but a Figure and should be placed in Supplementary Information.

Response: This Table has been modified to Figure 1 and simplified to better reflect how each processing step affected the color of the pesto product. Given suggestions provided by other reviewers, we decided to leave it in the main body of the manuscript.

Up to this point of the manuscript, the study has been focused on organoleptic and nutritional components’ aspects of kale pesto after sous-vide and/or blue light irradiation. However, there is no sensory panel evaluation to assess changes and acceptability/rejection of the product after processing, and it would be desirable to include these assays if it is intended to put the product into the market.

Response: Thank you for bringing this to our attention. We are aware that this study lacks sensory evaluation and consumer acceptability. Hence, we highlighted their necessity in the conclusions. Line 509.

L311-320 The hypothesis of VBNC cells are not sufficiently supported with the presented results. Please reconsider this fragment but I would recommend to delete it.

Response: This fragment has been reconsidered and rewritten, thank you. Line 327.

L321-329 This section has no discussion and must be added.

Response: This comment has been addressed. Please see the new paragraph. Lines 366-376.

MATERIALS AND METHODS

L350 Description column in Table 4 should be placed in the main body of the manuscript.

Response: Table 4 has been removed and all descriptions inserted into section 4.1.

L360 30 s vortex …. 5 min of centrifugation. Please uniform the same unit format.

Response: Unit format has been uniformed.

L365-370 This section should be merged into the appropriate sections of Materials and Methods.

Response: The section has been merged into sections 4.3 and 4.4.

L363 How was this concentration measured?

Response: This is an approximate concentration. Line 413.

L386 Give some details about the preparation of the working solutions.

Response: Details have been included, thank you for this suggestion. Lines 433-438.

L397 Give CIE full name.

Response: A full name has been inserted. Line 449.

L411-424 No information regarding the preparation of the bacterial stock cultures, working cultures, the method to adjust the optical density of the culture and the corresponding value in CFU/mL of the Abs600 = 0.8 AU.

Response: Detailed information has been provided. Please see section 4.6.

Reviewer 2 Report

Comments and Suggestions for Authors

The manuscript entitled "Improvement of quality attributes of turmeric-enriched kale pesto using innovative blue light-assisted minimal sous-vide processing" has novelty and well written. However, it needs some improvements as follows:

-Line 24: Please elaborate the term h° at first time use.

-Section 3.4: Na2CO3 à Na2CO3, Line 376: min. àminute.

-Section 3.5 needs a reference.

-Section 3.7: Please mention the source of L. monocytogenes strains. Also, mention why you used a cocktail of three strains instead of an individual strain. A mixture of strains may result in the self-destruction of bacteria during storage days.

-Line 416: 108 à 108

-Table 1: The antioxidant activity of kale pesto increased after all the treatments. Sometimes, it doubled or tripled. Although the authors tried to validate that, I think more vivid discussions with reasoning are necessary.

-Table 3: Please indicate 'the meaning of different numbers and highlighted spots' in the figure title or footnote for better readability for the readers.

- Since authors used turmeric as an antibiotic agent which has a very strong color and flavor, a sensory analysis of the products would give a better idea about the suitability of using curcumin in kale pesto.

-Please add a Conclusion Section.

Author Response

The manuscript entitled "Improvement of quality attributes of turmeric-enriched kale pesto using innovative blue light-assisted minimal sous-vide processing" has novelty and well written. However, it needs some improvements as follows:

-Line 24: Please elaborate the term h° at first time use.

Response: This has been addressed. Please see Line 22.

-Section 3.4: Na2CO3 à Na2CO3, Line 376: min. àminute.

Response: These have been corrected, thank you.

-Section 3.5 needs a reference.

Response: A reference has been added. Line 429.

-Section 3.7: Please mention the source of L. monocytogenes strains. Also, mention why you used a cocktail of three strains instead of an individual strain. A mixture of strains may result in the self-destruction of bacteria during storage days.

Response: Thank you for bringing this to our attention. Detailed information has been provided in section 4.6.

-Table 1: The antioxidant activity of kale pesto increased after all the treatments. Sometimes, it doubled or tripled. Although the authors tried to validate that, I think more vivid discussions with reasoning are necessary.

Response: We have carefully addressed this comment. Please see the paragraphs of the discussion that have been elaborated in red (section 3.1).

-Table 3: Please indicate 'the meaning of different numbers and highlighted spots' in the figure title or footnote for better readability for the readers.

Response: According to this comment and comments from the other reviewers, Table 3 has been modified to Figure 1. Now, we think it better illustrates the reconstructed colors of the pesto.

- Since authors used turmeric as an antibiotic agent which has a very strong color and flavor, a sensory analysis of the products would give a better idea about the suitability of using curcumin in kale pesto.

Response: Thank you for bringing this to our attention. We emphasized the necessity of conducting sensory analysis in the conclusions.

-Please add a Conclusion Section.

Response: Conclusions have been added. Lines 502-512. Thank you.

Reviewer 3 Report

Comments and Suggestions for Authors

Dear Authors

The study evaluated the effect of addition of turmeric and blue light-assisted sous vide on the quality properties of kale pesto. The topic of the study is interesting regarding employing hurdle concept to improve product safety and quality.

Overall comments:

Ther was no conclusion the body of the manuscript

Results and discussion section should be improved. It is better to describe the results firstly in consistent with tables and figures, then followed by specific discussion for each category of results. The authors started with general discussion for the results then followed by description.

Specific comments

The full name of measuring unit of Total Phenolic Content (TPC) in the abstract should be written

Line 21: the activity level, level of what?

Line 33-34: The phrase is vague. You have to restructure the phrase to impart clear meaning

Line 34: remove extra bracket

Line 75-78: The phrase is vague. You have to restructure the phrase to impart clear meaning

Line 97: I did not understand what did the authors mean with “affects the quality of green vegetables the least.”

Line 416: the count should be corrected to the power I think not 108 it is 108

Line 412: The L. monocytogenes strains used in this study were ATCC 15313, ATCC 19112, and 412 ATCC 19115, why this cocktail was used?

Results and discussion

This part should be organized. The results at first should be presented and described followed by specific discussion.

The CIELAB color space conversion to RGB color model in Table 3 has not been employed in the discussion. The author should explain the importance of such model in the context of the obtained results. Moreover, the authors should indicate the meaning of the numbers below figures of color space

Line 233: why authors used the results of previous study about ΔE values of packaged sliced cheese. The product is totally different from pesto

Line 269: delete “2017” from citation.

Figure 1 a. How the authors explain the increase in cell counts with turmeric in blue light treatment.

And how the author can explain that the addition of turmeric in sous vide did not show antimicrobial activity if compared to sous vide without turmeric

Author Response

The study evaluated the effect of addition of turmeric and blue light-assisted sous vide on the quality properties of kale pesto. The topic of the study is interesting regarding employing hurdle concept to improve product safety and quality.

Overall comments:

Ther was no conclusion the body of the manuscript

Response: We have addressed this comment and conclusions have been added. Lines 502-512. Thank you.

Results and discussion section should be improved. It is better to describe the results firstly in consistent with tables and figures, then followed by specific discussion for each category of results. The authors started with general discussion for the results then followed by description.

Response: Thank you for bringing this to our attention. Results and discussion have been improved accordingly.

Specific comments

The full name of measuring unit of Total Phenolic Content (TPC) in the abstract should be written

Response: This comment has been addressed.

Line 21: the activity level, level of what?

Response: Line 20: antioxidant activity level. Thank you.

Line 33-34: The phrase is vague. You have to restructure the phrase to impart clear meaning

Response: The sentence has been rewritten. Line 32.

Line 34: remove extra bracket

Response: The bracket has been removed, thank you.

Line 75-78: The phrase is vague. You have to restructure the phrase to impart clear meaning

Response: The phrase has been rewritten, thank you. Lines 95-99.

Line 97: I did not understand what did the authors mean with “affects the quality of green vegetables the least.”

Response: We apologize for the confusion. The sentences have been rewritten. Lines 103-106.

Line 416: the count should be corrected to the power I think not 108 it is 108

Response: This has been corrected, thank you.

Line 412: The L. monocytogenes strains used in this study were ATCC 15313, ATCC 19112, and 412 ATCC 19115, why this cocktail was used?

Response: Thank you for bringing this to our attention. Detailed information has been provided in section 4.6.

Results and discussion

This part should be organized. The results at first should be presented and described followed by specific discussion.

Response: We have addressed this comment and organized results and discussion as suggested.

The CIELAB color space conversion to RGB color model in Table 3 has not been employed in the discussion. The author should explain the importance of such model in the context of the obtained results. Moreover, the authors should indicate the meaning of the numbers below figures of color space

Response: The meaning of the numbers below figures of color space has been provided in the Suppl. Table 1. The description for the model has also been provided into the body of the manuscript. Lines 167-178.

Line 233: why authors used the results of previous study about ΔE values of packaged sliced cheese. The product is totally different from pesto

Response: A comparison to sliced cheese has been removed.

Line 269: delete “2017” from citation.

Response: This has been deleted, thank you.

Figure 1 a. How the authors explain the increase in cell counts with turmeric in blue light treatment.

Response: We have provided an explanation. Lines 198-199.

And how the author can explain that the addition of turmeric in sous vide did not show antimicrobial activity if compared to sous vide without turmeric

Response: Turmeric in sous-vide did show antimicrobial activity in fresh pesto: When turmeric was added, the reduction was 1.8 log CFU/mL, indicating a highly antimicrobial effect. Line 196. Subsequently, all cell counts declined, indicating that refrigerated storage time reinforced the observed effects, particularly for turmeric activated by blue light (Figure 2b). Line 203. This result suggests that blue light was a more prominent factor for antimicrobial activity to occur.

Round 2

Reviewer 2 Report

Comments and Suggestions for Authors

The authors responded to all the comments and improved or provided rebuttals accordingly. Now it can be accepted for publication. 

Reviewer 3 Report

Comments and Suggestions for Authors

No comments